# Deep Learning of Intrinsically Motivated Options in the Arcade Learning Environment

## Abstract

In Reinforcement Learning, Intrinsic Motivation motivates directed behaviors through a wide range of reward-generating methods. Depending on the task and environment, these rewards can be useful, might complement each other, but can also break down entirely, as seen with the *noisy TV* problem for curiosity. We therefore argue that scalability and robustness, among others, are key desirable properties of a method to incorporate intrinsic rewards, which a simple weighted sum of reward lacks. In a tabular setting, Explore Options let the agent call an intrinsically motivated policy in order to learn from its trajectories. We introduce Deep Explore Options, revising Explore Options within the Deep Reinforcement Learning paradigm to tackle complex visual problems. Deep Explore Options can naturally learn from several unrelated intrinsic rewards, ignore harmful intrinsic rewards, learn to balance exploration, but also isolate exploitative and exploratory behaviors for independent usage. We test Deep Explore Options on hard and easy exploration games of the Atari Suite, following a benchmarking study to ensure fairness. Our empirical results show that they achieve similar results than weighted sum baselines, while maintaining their key properties.

## 1 Introduction

In Reinforcement Learning (RL), an agent is sequentially given states and needs to perform actions in order to maximize obtained extrinsic rewards $r^e$. The agent is therefore deeply tied to the reward signal, and tends to fail when said signal is sparse or noisy. When the environment is very complex or high-dimensional, it is desirable for the agent to explore in a *directed* way (Thrun, 1992), i.e. explicitly looking for new knowledge and experiences. One of the most common ways to generate such task-independent, directed behaviors is through *intrinsic motivation* (IM), i.e. an alternative reward signal $r^i$ to spur curiosity and entice behavior exploration (Oudeyer & Kaplan, 2009; Schmidhuber, 2010). IM biologically refers to the natural tendency of organisms to explore.

One of the most common benchmarks for Deep RL agents has been the Arcade Learning Environment (ALE, Bellemare et al. (2013)), consisting of Atari video-games. In order to solve the most challenging, so-called *hard-exploration* games of the domain (Bellemare et al. (2016)), state-of-the-art Deep RL methods have integrated IM in complex learning mechanisms, and finally managed to overcome human-level play in all 57 games (NGU and Agent57, Badia et al. (2020b;a)). However, these methods still fundamentally rely on a Weighted Sum (WS) of rewards $r_t = r_t^e + \beta r_t^i$, for a very well-chosen and complex IM reward. So while the types of behaviors that we can extract with IM keep on expanding, we are still ultimately relying on a single signal to help exploitation. Instead, we might want to benefit from different and complementary intrinsic signals (Matusch et al., 2020). In this paper, we refer to this challenge as **IM Incorporation** (IM-Inc). We extract several key desirable features of IM-Inc methods, including scalability, robustness and generality.

In a tabular setting, Explore Options (EO, Bagot et al. (2020)) have been proposed as an alternative to the a weighted sum of rewards. The agent is divided into the Exploiter, trained exclusively with the extrinsic reward, and the Explorers, trained exclusively with the intrinsic rewards. The Exploiter can call any Explorers through *options* (Sutton et al., 1999), i.e. additional actions, to explore for a fixed amount of time. However, because of their tabular nature, Explore Options are inadequate for function approximation; but it is a crucial element to allow the agent to generalize the option over states and effectively learn to balance exploration. We revise Explore Options into *Deep Explore*

*Options* (DeepEOs), a new method for combining intrinsic and extrinsic reward signals in Deep RL. We introduce several key changes to usual IM approaches, and showcase their performance in the ALE. To provide fair and controlled comparisons, we match the algorithm and hyperparameters used in a benchmarking study (`benchmark`, Taiga et al. (2019)) on IM.

The contribution of our work is fourfold:

- We revise Explore Options within Deep RL to propose Deep Explore Options as a strong alternative to a weighted sum when using intrinsic rewards, extending the `benchmark`.
- We empirically show that, unlike a weighted sum of rewards, DeepEOs can learn from several IM rewards at once, and ignore harmful signals.
- We empirically show that DeepEOs can extract the exploiting behavior, while learning meaningful and potentially transferable exploratory behaviors.
- We provide a study of methods to *combine* intrinsic and extrinsic rewards. We propose several key desirable properties of such methods, and place several existing works within this framework, including DeepEOs.

In Section 2, we provide background to build Deep Explore Options. Next, we introduce our method and discuss the introduced elements in Section 3. In Section 4, we provide experiments in MiniGrid to build intuition, then in Atari following the `benchmark`. We go over existing work in the field in Section 5. Finally, in Section 6, we provide a study of methods to combine intrinsic and extrinsic rewards.

## 2 BACKGROUND AND EXPLORE OPTIONS

### 2.1 REINFORCEMENT LEARNING, OPTIONS, MOTIVATION

We use the standard RL setting (Sutton & Barto (2018)), modelling the environment as a Markov Decision Process $(\mathcal{S}, \mathcal{A}, \mathcal{R}, p, \gamma)$ where $\mathcal{S}$ is the set of states, $\mathcal{A}$ is the set of actions, $\mathcal{R} \subset \mathbb{R}$ is the set of rewards, $p : \mathcal{S}, \mathcal{A}, \mathcal{S}, \mathcal{R} \to [0, 1]$ is the dynamics function, and $\gamma$ is the discount factor. The goal of RL is to maximize the expected sum of discounted rewards from any starting state.
Options (Sutton et al. (1999)) refer to temporally extended actions. An option is defined as a triple $(\mathcal{I}, \pi, \beta)$, where $\mathcal{I} \subset \mathcal{S}$ is the option's initiation set, i.e. states in which the option can initiate; $\pi$ is the option's policy; and $\beta : \mathcal{S} \to [0, 1]$ is the option's termination condition.
We assume one or several intrinsic reward functions $f^{ir}(s, a, s') = r^i$ to generate a reward that we are interested in learning from. These can attempt to motivate directed exploration behaviors, but also any other behavior in the environment. An overview of the literature populating the field can be found in Section 5.

### 2.2 EXPLORE OPTIONS

Explore Options (`EO`, Bagot et al. (2020)) have been introduced as an alternative to a weighted sum of rewards. The method consists in decoupling the Agent into an Explorer, trained with the intrinsic reward $r^i$, and an Exploiter, trained with extrinsic reward $r^e$. Switching from Exploiter to Explorer is done through the Explore Option, which the Exploiter can use at any time to let the Explorer act for a fixed amount of steps $c_{switch}$. Within the options framework, the $j^{th}$ Explore Option $o_j$ is therefore defined as $\langle \mathcal{S}, \pi_j, \beta_{c_{switch}} \rangle$, where the initiation set $\mathcal{S}$ is the entire state space, the option policy $\pi_j$ is the Explorer policy trained with intrinsic reward $r^{i,j}$, and the termination function $\beta_{c_{switch}}$ deterministically interrupts the option call after $c_{switch}$ steps. Explore Options have only been introduced in a tabular setting. However, by design they only make sense in a function approximation setting: the option learning requires generalization in order to call the option in states where little is known, and therefore directed exploration is required. Function approximation also allows for parameter sharing, and potentially generalization across tasks.

## 3 DEEP EXPLORE OPTIONS

Fig. 1 gives an overview of the general Deep Explore Option framework, of which we go into more detail in the following subsections.

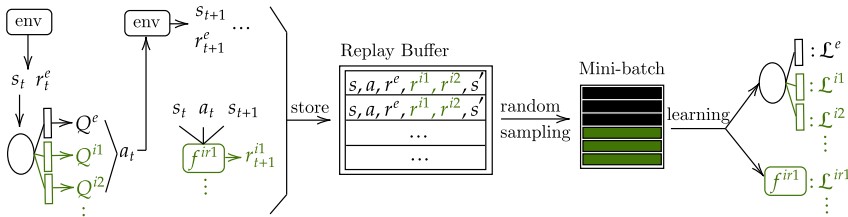

Figure 1: Deep Explore Option framework. During interactions, the state is passed through the DeepEO agent and either an Explorer (green) or the Exploiter acts according to the Explore Option. All transitions, enhanced with intrinsic rewards, are stored in the replay buffer, and sampled uniformly into a mini-batch, regardless of which agent had gathered them. The learning is therefore done off-policy on all transitions of the buffer.

### 3.1 CORE DIFFERENCES WITH USUAL INTRINSICALLY MOTIVATED AGENTS

**Option Learning**    A weighted sum of rewards relies on the IM rewards themselves guiding a dual exploiting-exploring agent towards relevant parts of the space. Instead, our IM is used exclusively to train an independent agent, that might be called through an Explore Option. We therefore rely on *off-policy learning*, i.e. the Explorer showing interesting transitions to the Exploiter, rather than merging the two behaviors – avoiding a complex $\beta$ tuning. Using the option has the benefit that the Exploiter can *learn* when to explore, e.g. as a default behavior when no rewards are known.

**Multiple Intrinsic Rewards**    Intrinsic reward functions $f^{ir}$ are often based on heuristics and intuition regarding what signal would motivate the agent to produce the most interesting and beneficial behaviors. The range of possible behaviors produced by Intrinsic Motivation is as wide as the task space, and widens with the complexity of the environment. Therefore, as we will further discuss in Section 6, and as recommended by Matusch et al. (2020), it is desirable to learn from several intrinsic reward signals. Due to the learning of a single agent per reward in DeepEOs, any off-policy algorithm guarantees the optimization of the intrinsic reward, and thus ensures scalability. The only practical issue with such approaches that learn different agents is if the scarce relative interacting of all agents begets the necessity of Offline RL (Fujimoto et al., 2019; Levine et al., 2020)).

**Auxiliary Tasks**    Sharing parameters of the model with auxiliary tasks in RL has been shown to provide a denser learning signal and accelerate learning, by building an environment-aware representation (Jaderberg et al., 2017; Lample & Chaplot, 2017). However, the learnt prediction and control models are rarely used beyond enriching the loss. We consider the learning of a fully intrinsically motivated policy or value function as an auxiliary task, but still use the learnt policy in practice. We therefore construct a feature representation that we branch out into our different Agent heads (see A.2 for the precise Architecture). The now-shared visual representation therefore needs to encapsulate relevant elements for both our Explorers and Exploiter. This type of architecture is quite common when learning from multiple signals (Barreto et al., 2018), but very rarely so when dealing with IM, mainly because the WS is hard to scale. In our experiments, a shared architecture is $50\%$ lighter and faster than separate networks. Note that each head and independent parts of the same optimizer can adjust for their own reward scale. In addition, our multi-headed architecture allows for efficient swapping, isolation or resetting of control heads, meaning that adding an Explorer or switching tasks can be done while preserving both the Explorer behaviors and the visual representation. We combine the losses through $\mathcal{L} = \mathcal{L}^e + \lambda \sum_j \mathcal{L}^i j$, with $\lambda = 1$, which required no tuning. $\lambda$ is not to be confused with the $\beta$ from WS, since each head adjusts for their own reward scales, meaning that $\lambda$ does not carry the heavy duty of reward balancing.

### 3.2 BACKBONE & TRAINING

**Benchmarking Study**    Experimental protocols in IM tend to be chaotic, with new approaches usually introduced on arbitrary environments and algorithms, and only compared to a non-IM baseline. Taiga et al. (2019) (`benchmark`) provides an experimental setup to fairly compare IM methods in the ALE, using the WS scheme. It uses the Dopamine framework (Castro et al., 2018),

---

**Algorithm 1:** Agent Loss Computation

---

**Input:** Buffer $\mathcal{B}$, Exploiter $Q^e$, $J$ Explorers $Q^{ij}$ and associated reward functions $f^{irj}, \forall j$

1   Uniformly **sample** batch $\left(s, a, r^e, r^{i1}, r^{i2} \cdots, s', done\right) \sim \mathcal{B}$       `// may contain options`

2   $option\_mask = a$ **in** $options$               `// mask of actions that are options`

3   $\mathcal{L}^e = $ **compute_loss** $\left(Q^e; s, a, r^e, s', done\right)$

4   **for** $j$ **in** $1 \cdots J$ **do**

5      $\mathcal{L}^{irj} = $ **compute_loss** $\left(f^{irj}; s, a, s'\right) *option\_mask$        `// mask out options`

6      $\mathcal{L}^{ij} = $ **compute_loss** $\left(Q^{irj}; s, a, r^{ij}, s', done = False\right) *option\_mask$

7   $\mathcal{L} = \mathcal{L}^e + \lambda \sum_j \mathcal{L}^{ij} + \sum_j \mathcal{L}^{irj}$

**Output:** Total Loss $\mathcal{L}$

---

which provides an implementation of the Rainbow agent (Hessel et al., 2018) with its 3 majors improvements on DQN (Mnih et al., 2015): Prioritized Experience Replay (Schaul et al., 2016), n-step returns, and Categorical DQN (C51, Bellemare et al. (2017)). We match this setting and all hyperparameters using the DeepRL PyTorch implementation (Zhang, 2018).

**Intrinsic Reward Functions**    The `benchmark` compares several intrinsic reward functions; for our first IM function we use Random Network Distillation (RND, Burda et al. (2019b)), as it is the simplest, most used method, and is currently part of state-of-the-art methods (Badia et al., 2020b;a). RND uses a fixed, randomly initialized *target* network $f : \mathcal{S} \rightarrow \mathbb{R}^m$ and a learnt *predictor* network $\hat{f} : \mathcal{S} \rightarrow \mathbb{R}^m$. The method generates an intrinsic reward $r^{i1} = f^{ir1}\left(s, a, s'\right) = ||\hat{f}\left(s'\right) - f\left(s'\right)||_2^2$ by distilling the target into the predictor, using the loss as motivation signal. This will therefore lead the agent towards less-visited states, that still show a high distillation error. In order to keep our return estimates within C51's usual $[-10, 10]$ range, we scale the RND reward by $\eta = 0.01$. This is not to be confused with the $\beta$ from `WS`, as this $\eta$ is vastly easier to tune: it is only meant to keep the RND returns within a reasonable scale and never meant to balance different reward signals.

In order to study a multi-IM setting, we use another intrinsic function that is trivial to implement: Optimistic Initialization (Machado et al., 2015). It can be obtained through a simple constant negative reward, which will push the agent off visited grounds. The original paper balances for the lost rewards at termination, but our purely intrinsically motivated agent is not provided a terminal signal, and can therefore simply be fed a constant $r^{i2} = -0.025$. In this scenario, this agent is a close cousin of the DORA method of exploration (Fox et al., 2018).

Since virtually all IM rewards are based on heuristics, they are bound to break in certain scenarios and environments. A common example of such failure is the *noisy TV problem* (Burda et al., 2019a): an agent trying to find sources of surprise will be mesmerized by unpredictable noise. In order to test our method for such a breakdown, we implement a third intrinsic reward function $r^{i3} \sim \mathcal{N}\left(0, 1\right)$ that samples rewards randomly in each mini-batch. An agent trained with this will be chaotically attracted to random parts of the state-action space.

**Agents and Option Training**    We build our multi-headed agent with an Exploiter head and several Explorer heads. We add actions to the Exploiter, corresponding to the `EO`s. When the option $o_j$ is called, the corresponding Explorer $j$ acts for a fixed $c_{switch}$ steps before giving control back to the Exploiter. This transition $(s, a, r, s') = (s_t, o_j, R, s_{t+c_{switch}})$, with $R = \sum_{k=0}^{c_{switch}-1} \gamma^k r_{t+k+1}^e$, is added to a unique buffer along the rest of the $n$-step transitions, regardless of the current actor. We train all agents simultaneously, off-policy, using the same uniformly sampled buffer data, masking the option transitions for the Explorers, as explicited in Algorithm 1. The Explorers are not provided the `done` signal, since it might leak information about the task, rush agents with negative rewards, and interfere with exploration (Burda et al., 2019a).

In the Dopamine implementation of Rainbow, $\epsilon$-greedy is the main strategy used for exploration. Keeping this unchanged for options would mean enforcing $c_{switch}$-long exploration phases when the options are randomly called, leading to much more exploration than $\epsilon$-greedy intends. Instead, we note that $\epsilon$-greedy means allocating a fraction $\epsilon$ of the agent's time to exploring. In order to preserve this idea, we enforce our Explorer agents to occupy $\epsilon/2$ of the agent's time, while our Exploiter otherwise follows an $\epsilon/2$-greedy exploration strategy.

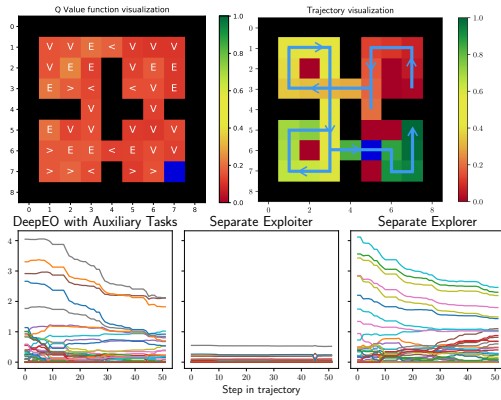

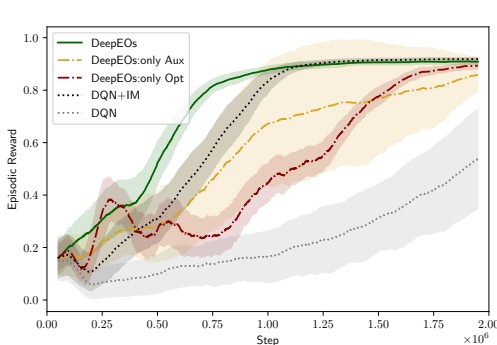

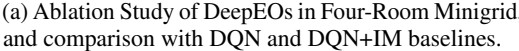

(a) Ablation Study of DeepEOs in Four-Room Minigrid, and comparison with DQN and DQN+IM baselines.

(b) Visual intuition for DeepEOs: [top left] early policy, after 250k steps, where E represents a DeepEO call (blue: goal, agent orientation: down); [top right] an Explorer trajectory; [bottom] Learnt features with and without auxiliary task learning, when following the Explorer trajectory.

Figure 2: Experiments in MiniGrid.

## 4 EXPERIMENTS

### 4.1 MINIGRID INTUITION

We start by building intuition for DeepEOs in MiniGrid (Chevalier-Boisvert et al. (2018)), a 2D visual gridworld. We use a classic four-room environment with actions {forward, left, right} and fully-observable inputs. We implement a simple first-cell-visit intrinsic reward $f_{ir}(s, a, s') = \mathbf{1}_{s' \notin \tau}$, which we use to train our DeepEO agent based on the DQN algorithm (Mnih et al. (2015)). We also use it to train a weighted sum DQN+IM agent as a baseline, along with an untouched DQN agent. We design an ablation study of DeepEOs, removing either the Option or the Auxiliary Task learning (Section 3.1) to leave only the other. The results can be found in Figure 2a. We can see that without ablations, DeepEOs perform better than the baselines. The ablation studies show that both the option and the auxiliary tasks are beneficial for learning in this scenario, with the auxiliary task rather improving early performance, and option usage rather late performance. In Figure 2b, we visualize several key components of the method. First, we show the Exploiter policy early in training, after $250k$ steps. We can see that the agent quickly learns to call the Explorer (action E) in states distant from the goal. Next, we show an Explorer trajectory towards the end of training, to confirm that it achieves a strong and independent covering policy. Finally, we study the representation with and without auxiliary task learning, by observing the network's latent representation during the Explorer trajectory. We freeze the agent's position in the trajectory to only let the coverage map change. We can see that the Exploiter alone (similar to a DQN agent) essentially ignores the coverage map. However, the Explorer needs it to act, so its features are sensitive to it. Combining both networks therefore forces the Exploiter's representation to contain information about the coverage map, which is useful for performance (Figure 2a). More information about all experiments in this subsection can be found in Appendix A.1.

### 4.2 ATARI EXPERIMENTAL PROTOCOL

We now evaluate Deep Explore Options on the Arcade Learning Environment, with RND and Optimistic Initialization Explorers, with the same games, core algorithm, architecture and hyperparameters as the Rainbow implementation from the `benchmark`. Architecture and full experimental details can be found in Appendix A.2. We use two baselines provided in the `benchmark`: Rainbow with $\epsilon$-greedy (`Rainbow`) and Rainbow with an RND intrinsic reward using `WS` (`Rainbow+RND`). We refer to our method as `DeepEO`. In order to test our exploitation behavior, we evaluate our Exploiter under an $\epsilon$-greedy *without options* every million frames; referred to as `Exploiter`.

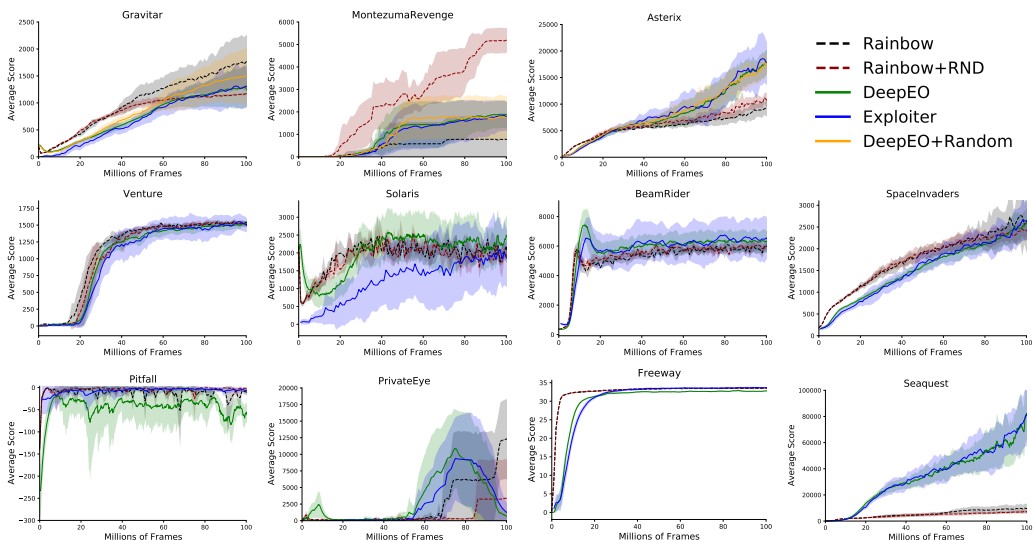

Figure 3: Experiments on the Atari Suite. Two left columns: hard-exploration games. Two right columns: easy-exploration games. The blue curves represents the $\epsilon$-greedy performance of the Exploiter without forced usage of the Explore Options. The orange curve represents the DeepEO in a robustness challenge: with a purely Random reward.

## 4.3 MAIN RESULTS

**Performance and Scalability** Our main empirical results are shown in Figure 3. We observe that our results are overall competitive with the baselines. Except for SOLARIS, the Exploiter always improves on `DeepEO` where option usage is forced. On PRIVATEEYE, our agents learns sooner, but therefore also unlearns sooner – this unlearning is common in this environment. The two most notable outliers are GRAVITAR, where `DeepEOs` do not manage to match `Rainbow`, and MONTEZUMA, where it does not manage to match `Rainbow+RND`. We address the latter in 4.4, but note that we always improve on the lower bound. In FREEWAY but most notably in PITFALL, the `DeepEO` performance lags behind the `Exploiter`. While this is expected in Freeway, since there is always a small chance to call the Explorers for $c_{switch}$ steps, the case of `Pitfall` is interesting: the Exploiter wants to do nothing to get a return of $0$, but both Explorers are attracted to interactions that eventually lead to negative rewards. The `Exploiter` successfully ignores them. This is a first hint that the Exploiter ignores harmful behaviors and extracts the exploiting behavior.

**Robustness** Following this remark on `Pitfall`, we study Robustness by introducing our third IM reward: the purely random $R_{i,3} \sim \mathcal{N}(0,1)$, simulating a signal that degenerates. This is represented as DEEPEO+RANDOM in the tuning games. We can see on Figure 3 again that this new intrinsic reward had absolutely no harmful consequence on our performance, matching `DeepEO` without a hitch. This shows that DeepEOs are perfectly robust to destructive IM rewards. For the sake of argument, we try a Weighted Sum of our 3 rewards and report the results in Appendix B, 6 as MULTI RAINBOW. As expected, a weighted sum of rewards is not scalable nor robust unless under heavy $\beta_j$ fine-tuning.

**Additional Experiments** We include two more experiments in Appendix B: first, a study on the $c_{switch}$ hyperparameter, where we find performance to be very robust to its changes. We choose $c_{switch} = 50$ in the experiments we display in the main text as the best value by a small margin. Next, we study naive PER prioritization for 2 agents, which we find does not benefit the agent; therefore we used random sampling for our DeepEO agent. We encourage future work to design algorithms to perform multi-agent prioritization in a similar context.

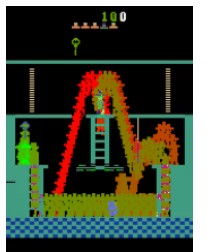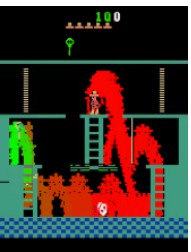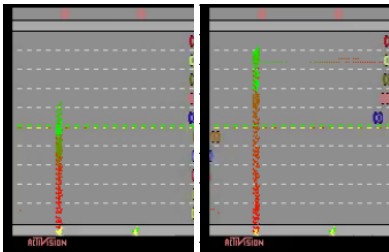

Figure 4: Visualization of some learnt behaviors of, respectively, the RND and Optimistic Explorers. Red to green respectively mean older to newer visits in a single episode. On MONTEZUMA (left), both agents learnt to reach the key, but the RND Explorer is much faster, while the Optimistic Explorer tends to jump more but lags on the ladder leading to the key. On FREEWAY (right), the RND Explorer is interested by the fast-lane cars in the middle, while the Optimistic Explorer likes the upper, less visited section.

## 4.4 LEARNT BEHAVIORS

Previous research has shown that temporally extended exploration can be a surprisingly efficient exploration method (Nachum et al., 2019; Dabney et al., 2021), even when said exploration is random or spamming. While this is a motivation for our work, it raises an important question - did our Explorers learn any relevant behaviors, or are we simply benefiting from observing several steps of random off-policy behavior? While Figure 2b gave a positive answer for MiniGrid, we now study Atari. We visualize the learnt behaviors in Figure 4. Redder shades indicate earlier timesteps in the episode, while greener represent most recent timesteps. In MONTEZUMA, both the RND and Optimistic Explorers learn to reach the key on their own, showcasing long-term planning. *We note the relative weakness of the independent policies compared to a full online PPO RNN agent* in the original RND paper. We believe improving the Explorer quality is an important axis for future improvement, and recommend studies on the types of algorithms that suit IM most, and on the impact of near-offline learning. These results are still motivating for Task Transfer: in Campos et al. (2021), the authors pre-train an exploratory behavior and use it as an option downstream. As they mention, this approach complements well with EOs, as they allow training of the exploration policies in tandem with exploitation.

## 5 RELATED WORK

**Exploration without IM** Methods that do not incorporate IM were state-of-the-art on Atari until recently. Such methods generally injected noise in the action or parameter space to explore. In Value-based methods, $\epsilon$-greedy (Mnih et al., 2015) randomly explores with some probability at each step, while Noisy Nets (Fortunato et al., 2018; Hessel et al., 2018) introduces a learnt noise in the parameters. In Policy-based methods, entropy maximization (Haarnoja et al., 2018) takes advantage of the stochastic policy formulation to motivate noisier policies.

**Intrinsic Motivation: Knowledge Acquisition and Skill Learning** IM (Aubret et al., 2019) is generally used for *knowledge acquisition* about the environment. Motivating exploration is by far the most common goal then, and is divided into *state novelty* (Bellemare et al., 2016; Ostrovski et al., 2017; Burda et al., 2019b), *prediction error* (Pathak et al., 2017) and *information gain* (Schmidhuber, 2010). Beyond knowledge acquisition, IM can also encompass *skill learning*, where the agent learns a set of temporally-extended actions to help solve the task, sometimes called *option-* or *skill-discovery*. The main body of work involves reaching states with key properties, for example through Proto-Value Functions Machado et al. (2017), algebraic graph connectivity (Jinnai et al., 2019a;b), the Successor Representation (Ramesh et al., 2019) or simply a measure of diversity (Eysenbach et al., 2019). These methods generally attempt to find a coherent set of options, so they do not try to incorporate other forms of IM exploration. In other words, Explore Options generalize these ideas to any set of intrinsically-motivated options, and could benefit from these methods as Explorers.

## 6 DISCUSSION: COMBINING REWARD SIGNALS

As explored in the previous section, the generation of useful, auxiliary reward signals under the form of intrinsic motivation has been an active subject in recent literature; we refer to this task as **IM Generation** (IM-Gen). However, addressing *how* to incorporate and learn from these reward signals has seen surprisingly less attention; we refer to this task as **IM Incorporation** (IM-Inc). In this section, we extract key desirable properties of IM-Inc methods. We then discuss the position of several state-of-the-art methods in this framework.

### 6.1 DESIRABLE PROPERTIES

We start by assuming the usage of several IM-Gen methods $f_{ir,j}(s, a, s') = R_{i,j}$, as a set of rewards that simply *might* be useful. The goal in introducing these is to *boost the present or future exploiting performance*. By *future*, we mean to cover Transfer and Lifelong Learning: learning representations and behaviors that will solve downstream tasks in similar environments. With this in mind, we find the following desirable properties for an IM-Inc method:

**Scalability**   It is not realistic to expect of a unique IM function that it should single-handedly maximally assist the exploiting behavior. IM-Gen methods generate all types of behaviors and data, and might complement each-other within a single task or over tasks. From Matusch et al. (2020): "we find that input entropy and information gain are similar objectives while empowerment may offer complementary benefits, and thus recommend future work on combining intrinsic objectives." In Ramakrishnan et al. (2021), the strongest exploration-based rewards change drastically with the environment and task. *An IM-Inc method should therefore be able to combine several IM rewards.*

**Robustness**   Following scalability, among the many rewards, some will be less effective, while some might even degenerate for portions of or entire tasks (e.g. *noisy TV* problem, Burda et al. (2019a)). *An IM-Inc method should be able to remain unaffected by harmful signals.*

**Behavior Assimilation**   IM, in essence, leads to a conflict of interest between exploitation and the IM behaviors. An IM-Inc method should therefore manage to assimilate these behaviors in a way that helps exploitation in all scenarios, and does not risk losing any of the relevant behaviors. A simple degenerating scenario is that an $r_{grab} + r_{run}$ agent could lose both independent behaviors to chaos.

**Generality**   To ensure convergence, IM-Gen methods often rely on some assumption on the rewards - for example that they will disappear in time, or that they cover exploiting behavior. However, this is vastly limiting in the types of rewards and behaviors it allows. Therefore, along with scalability, it should be effortless to introduce and swap in new IM-Gen rewards, regardless of their nature. *An IM-Inc method should be minimally restrictive on the rewards it allows, and let them swap in easily.*

**Transferability of Behaviors**   Most IM-Gen methods produce independent behaviors that are meaningful for several tasks. Learning such behaviors is therefore of great interest not only in the current task, but also for future ones, in task transfer scenarios under similar dynamics. *The ability to preserve and re-use these behaviors is thus of great relevance* to severely boost the learning of down-stream tasks, as done in Campos et al. (2021). We refer to this idea as *Behavior Transfer*.

**Representation Learning**   In a function approximation scenario, the learning of an IM policy always requires to understand the structure of the state and MDP to some extent, through some form of representation learning. Having the exploiting behavior benefit from this understanding through auxiliary tasks or additional inputs can strongly benefit performance. Even beyond this, and following transferability, *the ability to extract and re-use these learnt representation for down-stream tasks could lead to substantial gains in performance* (e.g. Gordon et al. (2019)). We refer to this recycling idea as Representation Transfer (Lazaric, 2012).

### 6.2 STUDY AND TABLE

We identify several methods that can categorize as IM-Inc from the literature. We organize them in Table 1 following the key properties we have formulated.

First and most obviously, a weighted sum of rewards, as done in the `benchmark`, is the most common IM-Inc method. The behavior assimilation method is through a merged exploiting/exploratory Agent that requires fine attention to $\beta$, but this makes it hard to scale and hard to swap in new rewards. It is also very restricting: if the intrinsic reward doesn't disappear in time, it will always harm the

| Method | Scalability | Robustness | Behavior Assimil. | Generality | Transfer Potential | Representation Learning |
|---|---|---|---|---|---|---|
| Weighted Sum | No | No | Merged | No | No | Merged |
| NGU | No | **Yes** | Merged | No | No | **UVFA** |
| Agent57 | No | **Yes** | Merged | No | **Yes** | No |
| Successor Features | **Yes** | **Yes** | **GPI** | No | **Yes** | **Aux. Tasks** |
| Explore&Exploit | **Yes** | No | Separate | **Yes** | **Yes** | No |
| **DeepEO** | **Yes** | **Yes** | Separate | **Yes** | **Yes** | **Aux. Task** |

Table 1: Comparison of methods to combine intrinsic and extrinsic rewards

exploiting behavior. Never Give Up (NGU, Badia et al. (2020b)), combines an intra and inter-episodic IM reward through a complex mix of scaling, bounding and eventual weighted sum: we see this reward as the limit of single-reward formulations. In NGU, a UVFA network (Schaul et al., 2015) allows to pass $\beta$ as input and thus fine-tune $\beta$ during training-time. This makes NGU robust to harmful signals. While the UVFA is beneficial for Representation Learning, it does not allow for efficient Representation Transfer since we cannot untangle the Exploiter and Explorer to isolate the second. Agent57 (Badia et al., 2020a) improves on NGU by further fine-tuning $\beta$ with a Multi-Armed Bandit, and replacing the UVFA by decoupled Explorer and Exploiter Value Functions for stability. This effectively enables Behavior Transfer of the learnt Explorer to a degree, but the Exploiter does not benefit from the Explorer representation anymore.

Successor Features (SF, originally Dayan (1993)) are a family of methods relevant here through the General Policy Improvement Theorem (GPI, Barreto et al. (2017) and Barreto et al. (2018)), allowing to extract the strongest available policy from any set of base policies. While this is ideal for Behavior Assimilation and Transfer, current SF methods require a very expressive set of base rewards in order to formulate any new reward as a linear combination of the basis. This assumption is yet to be proven viable in a general setting. Regarding Representation learning, Barreto et al. (2018) used an architecture similar to ours, sharing parameters in an auxiliary task manner.

Finally, we include with our DeepEO method along with its closest cousin, Explore&Exploit (Nachum et al., 2019). Explore&Exploit independently trains a purely IM Agent and samples the next Agent between Explorer and Exploiter every $c_{switch}$ steps. Due to this fixed sampling, this method is not robust[1], while DeepEOs can learn to ignore bad options (Section 4.3). Both methods are scalable and general (4.3), as guaranteed by the learning of a separate behavior per reward function (Section 3.1). Note that using separate behaviors is necessarily weaker than the GPI, by definition. The shared architecture of DeepEOs allows to benefit from the Explorer Representation (Section 4.1). Regarding Transfer, Campos et al. (2021) provide a method to efficiently transfer from learnt exploratory behaviors using $c_{switch} = 1$ for Exploiter-called Explore Options and a higher value for random option calls. They mention Explore Options as a complementary method to their work for training the Explorers in tandem with the Exploiter.

## 7 CONCLUSION

We have introduced a taxonomy for methods to combine intrinsic and extrinsic rewards in Reinforcement Learning (IM-Inc). We have extracted, as desirable properties of such methods: scalability, robustness, behavior assimilation, generality, and transferability of the behavior and representation. Most of these are not met by a weighted sum of rewards, the usual approach to combining rewards. We have revised Explore Options to Deep Explore Options, to call an intrinsically motivated agent through an option in a function approximation setting. To ensure fair comparisons, we followed a benchmarking study that uses hard- and easy-exploration games of the Arcade Learning Environment. We showed here that Deep Explore Options are a strong alternative to a weighted sum of rewards, with similar performances on most games. But in addition to performance, we have empirically demonstrated that DeepEOs possess most of the desirable properties of an IM-Inc method – in particular scalability, robustness, generality and representation learning.

---

[1]We should note that Explore&Exploit isn't meant as an IM-Inc method, but to study Hierarchical RL.

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

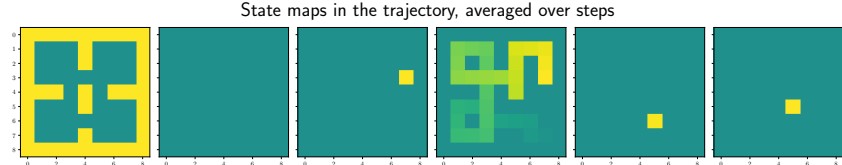

State maps in the trajectory, averaged over steps

Figure 5: Minigrid: One-hot encoded maps passed to the agent, here averaged over a covering trajectory, keeping the position fixed. In order: [walls, agent_x, agent_y, visits, goal1, goal2].

## A    IMPLEMENTATION & EXPERIMENTAL SETUPS

### A.1    MINIGRID EXPERIMENTS

For simplicity, the agent is passed a fully observable tensor of its state in the environment. In particular, to avoid requiring a memory, the state we use for Minigrid experiments contains one-hot encoded maps as [walls, agent_x, agent_y, visits, goal1, goal2]. The agent x and y correspond to its orientation in space, only passed at the cell the agent is occupying. The visit map marks as 1 each visited cell. The agent needs to ignore the last goal2 map.

For our feature-visualization experiment, we used a trajectory of 50 steps (top right of Figure 2b) and fixed the agent positions maps, leading to Figure 5: only the visitation map changes, here average over the trajectory.

We trained our DQN-based agents for 2 million steps, with $50k$ steps of random behavior, $\epsilon$ decreasing from 1 to 0.01 over $100k$ steps, network updates every 4 steps, target network updates every $10k$ network updates ($40k$ steps), a learning rate of 0.00025 with Adam optimizer, discount of $\gamma = 0.9$. The DQN+IM agent was fine-tuned to $\beta = 0.1$ as the best hyperparameter value. Our DeepEO agent used $c_{switch} = 50$.

### A.2    ATARI HYPERPARAMETERS & ARCHITECTURES

| Hyperparameter | Value |
|---|---|
| Discount factor $\gamma$ | 0.99 |
| Min history to start learning | 80K frames |
| Target network update period | 32K frames |
| Adam learning rate | $6.25^{-5}$ |
| Adam $\epsilon$ | $1.5^{-4}$ |
| Multi-step returns $n$ | 3 |
| Distributional atoms | 51 |
| Distributional min/max values | $[-10, 10]$ |
| $\epsilon$-greedy schedule | $1 \rightarrow 0.01$ over 1M frames |

Table 2: List of Rainbow hyperparameters used, taken from the `benchmark`

**Hyperparameters**    The list of used Rainbow hyperparameters can be seen in Table 2 above.

**Architecture**    The network architecture used is based on the Categorical DQN, as used in Dopamine (Rainbow without Dueling). The network has 3 Convolutional layers followed by 1 fully-connected layer, which we refer to as the "vision module", followed by a fully-connected output layer, which we call "control module". The Exploiter has one more action per Explorer. Each Explorer attaches a "control module" to the shared body "vision module".

**Experimental setup**    Following the `benchmark` and recommended practices (Machado et al., 2018), we use $\varsigma = 0.25$ sticky-actions and no termination on life loss.

Agents are trained over 5 seeds for up to 100 million frames on the 11 games selected by the `benchmark`. 6 of them are hard-exploration games, while 5 are easy-exploration games.

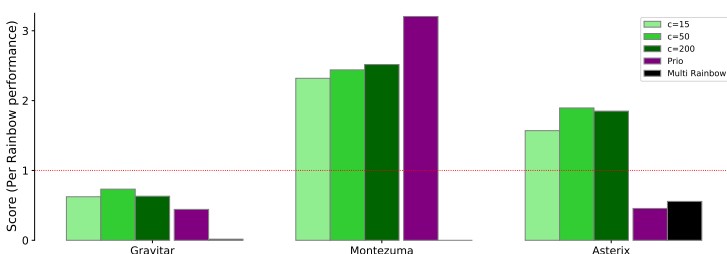

Figure 6: Additional experiments: $c_{switch}$ study; Prioritization; multi-reward Weighted Sum

# B    ADDITIONAL RESULTS

## B.1    $c_{switch}$ AND PRIORITIZATION STUDIES

We perform experiments on our three tuning games: GRAVITAR, MONTEZUMA and ASTERIX. We chose these because they are a mix or easy and hard-exploration that provide relatively smooth improvements with better methods. We could not match the set from the `benchmark` as it tunes on games outside the main set of 11.

$c_{switch}$ **hyperparameter**    In Bagot et al. (2020), the authors linked EO's $c_{switch}$ hyperparameter to WS' $\beta$, since both dictate the intensity of exploration in their respective approach. It was shown that $c_{switch}$ was much more robust and lead to overall better performances than any value of $\beta$ in a simple setting. We do not have access to the `benchmark`'s per-$\beta$ performance but only the range: $\beta \in \{0.05, 0.01, 0.005, 0.001, 0.0005, 0.0001, 0.0005, 0.00001, 0.000005\}$. We replicate this study in Atari by studying $c_{switch} \in \{15, 50, 200\}$. The results are available in Figure 6, normalized by the `Rainbow` performance. We find that $c_{switch}$ does not have a lot of impact in performance, with all values leading to very similar performances in all tuning games. $c_{switch} = 50$ did better on GRAVITAR and ASTERIX by a more convincing margin than $c_{switch} = 200$ did on MONTEZUMA. We therefore pick 50 as our main hyperparameter value.

**Prioritization**    Rainbow uses Prioritized Experience Replay (`PER`, Schaul et al. (2016)) to sample more interesting transitions from the buffer, where the notion of priority of a transition is its associated Rainbow loss. Since we now have at least two agents, and therefore two notions of priority, a naive adaptation of PER would lead to each agent sampling a mini-batch according to its own preferences. We implement this idea, but as can be seen in Figure 6, we find that it vastly under-performs against random sampling on both GRAVITAR and ASTERIX – the only one on which it was quite beneficial unsurprisingly being MONTEZUMA. A potential cause for this gap in performance could reside in the option transitions: their higher average loss scales might be interfering with the sampling process. We leave deeper investigations of such multi-objective prioritization to future work.

