# OpenReview forum: "Deep Learning of Intrinsically Motivated Options in the Arcade Learning Environment"
_ICLR.cc/2023/Conference — Submitted to ICLR 2023_

### Official Review · Reviewer_aMNX · 2022-10-20

**Confidence:** 5
**Correctness:** 1
**Technical Novelty And Significance:** 1
**Empirical Novelty And Significance:** 1
**Recommendation:** 1

**Clarity, Quality, Novelty And Reproducibility:**

The paper has almost no scientific novelty. Previously introduced methods are combined based on intuitive arguments, not supported by theoretical guarantees. The experiments are not well reported. The clarity of the paper is not satisfactory.

**Strength And Weaknesses:**

This paper proposes a basic deep RL extension of previously proposed Explore Options. This extension is very straightforward, as evinced by the very small description dedicated to it in the paper, and it does not have scientific significance or novelty. The methodological section investigates ways to combine previously introduced techniques to decouple the learning of intrinsic and extrinsic rewards, without any theoretical intuition of how these methods should be effective. The empirical section is not well reported, as the figures are mostly unreadable and poorly informative. In general, the writing of the paper is not satisfactory, with an abuse of terminology, e.g., "hard-exploration games" is not a common terminology, and overclaims.

**Summary Of The Paper:**

This paper extends the use of previously proposed Explore Options to deep Reinforcement Learning (RL). It tests different methods to incorporate intrinsic rewards, that go beyond the typically used approach of summing extrinsic and intrinsic reward signals.

**Summary Of The Review:**

The paper has major weaknesses counterbalanced by a few strengths. The paper claims that the proposed contribution is "fourfold"; this is concerning to me, since in my opinion the contribution of this paper is actually quite minimal. I recommend the author seriously reconsider this work and resubmit it only with substantial improvements.

---

### Official Review · Reviewer_u5M2 · 2022-10-23

**Confidence:** 5
**Correctness:** 1
**Technical Novelty And Significance:** 1
**Empirical Novelty And Significance:** 2
**Recommendation:** 1

**Clarity, Quality, Novelty And Reproducibility:**

# Clarity

I found the paper's motivation to be quite unclear, buried in the text, and took multiple passes to understand exactly what problem with current methods you are aiming to solve with DEO. In the intro, you mention that your approach provides scalability, robustness, and generality but none of these terms are defined until towards the end of the paper (e.g. scalable or robust with respect to what?). I suggest being more precise about these claims and defining such terms the first time they are mentioned (e.g. by bringing the core of section 6.1 in the intro).

I also found Section 6.1 to be quite confusing since many of the claims are not supported by the experiments and read like future work, rather than strengthening the current work. On the other hand, the first part of this section seems to contain motivation for the approach and current work so would be better placed in the introduction.

The related work is quite short and the paper doesn't cite many relevant works. Frankly, there are too many to list here but recommend the authors to do a thorough review of the exploration and intrinsic motivation literature and add more references.

# Quality

The paper makes multiple claims which are not well-supported by the current experiments.

**Scalability**. In section 6.1, you talk about the importance of making use of complementary exploration bonuses, but I don't think the current results show this convincingly. In addition, typical methods that use a weighted sum are also able to combine multiple intrinsic rewards. So how do we know your approach is better at combining different bonuses? Can you run experiments with complementary intrinsic rewards (e.g. info gain and empowerment) for both your approach and the baselines? It would also be better to use more common intrinsic rewards such as ICM [1] or PseudoCounts [2] rather than or in addition to the optimistic initialization.

**Robustness**. Another core claim is that your method is unaffected by harmful intrinsic rewards, but you never demonstrate that typical methods suffer from this. Running experiments to verify this claim could further improve the quality of the paper.

**Generality**. Can you show that indeed your approach learns to not use the harmful option throughout training?

You also discuss behavior assimilation, transferability of behaviors, representation learning, none of each are supported with experiments and some seem out of scope for the current paper which focuses on exploration in singleton environments, rather than transfer to new tasks and lifelong learning. While indeed they might be strengths of the proposed approach, I would only mention them briefly in the discussion or appendix since they are not supported by your experiments. To me, these seem like they could be part of a follow-up paper focused on transfer learning. Alternatively, if you want to make them central points, you need to provide further experiments supporting them.

**Analysis**. It would be useful to visualize the behavior of each option, and analyze how they are used by the high-level policy throughout training.

In section 6.1 you are comparing DEO with other approaches qualitatively, but not quantitatively, which seems somewhat dubious. While DEO in theory has all these advantages, these aren't empirically verified in the paper, so the claims should be toned down and make clear that further investigations are needed to see if this translates into practical gains.


# Novelty

As mentioned, DEO seems to be a fairly trivial extension of EO. The authors don't provide much detail on how it differs from EO, other than using function approximation to extract features from high-dimensional observations, which is standard practice in deep RL.

However, I wouldn't hold this against acceptance if the other issues are resolved and in particular.


# Reproducibility

I cannot see the code for this paper, but the authors provide a reasonable amount of detail regarding their experiments and implementation. However, I don't think the details are enough to easily reproduce the results without access to the code. It would be great if the author can comment on their plans of open sourcing the code in order to ensure reproducibility.


**Strength And Weaknesses:**

# Strengths:
1. developing a general exploration method that can work across many environments is an important problem
2. the core idea of having separate policies that explore using different intrinsic rewards, along an exploitation policy, makes sense
3. the proposed method is simple and intuitive

# Weaknesses:
### 1. Multiple claims are not well-supported by the experiments
- You claim that DEO can better benefit from multiple types of exploration but never compare it with an equivalent Rainbow that uses the same number of intrinsic rewards. I suggest comparing DEO+RND+OI with Rainbow+RND+OI.

- You also never demonstrate that the exploration bonuses used have complementary benefits although you mention that as a strength of DEO. Can you include DEO+OI, DEO+RND in the comparison and ideally other variants like DEO+ICM, DEM+RND+ICM, DEO+RND+ICM+OI. These also need to be compared with their Rainbow counterparts. Other exploration bonuses are fine too as long as they are orthogonal and commonly used.

- You claim that DEO is more robust to harmful exploration bonuses but never compare DEO+Random with Rainbow+Random.

- You claim that DEO learns transferable exploratory behaviors but you never validate this. If you want to make this claim, you need to evaluate the learned policies on new tasks and show that a policy using them learns faster than without them.

- You claim that DEO extracts the exploitative behavior but this is not supported by experiments. Can you show how the agent selects the options throughout training and that towards the end it converges on always selecting the exploitative one?

### 2. Limited empirical evaluation, not enough baselines or ablations
- You only use one common intrinsic reward RND but claim that your approach can benefit from complementary intrinsic rewards, much more than standard approaches. I suggest running experiments with other common exploration bonuses such as ICM [1] or PseudoCounts
[2], and comparing DEO with RND-only, DEO with ICM-only, DEO with both, as well as all the equivalent Rainbow

- In general, policy-optimization algorithms like PPO or IMPALA tend to benefit more from intrinsic rewards than value-based algorithms like DQN or Rainbow. In fact, it is quite odd to use such exploration bonuses with DQN / Rainbow. Typically, value-based methods benefit more from things like NoisyNet [3], Bootstrapped DQN [4], ez-greedy [5] etc. Hence, I wonder if the (lack of?) benefits from DEO is mostly coming from the fact that the baselines are weak. It's possible that DEO actually improves more over the baselines when implemented on top of  a more appropriate base algorithm Thus, I recommend applying DEO to policy-optimization algorithms like PPO or IMPALA and comparing it with its weighted-sum equivalent.

- You claim that $\eta$ is vastly easier to tune and don't really consider it a HP. However, this claim isn't well supported. You need to show a sensitivity analysis with respect to $\eta$ and $\beta$ for the two approaches in order to evaluate whether DEO is indeed less sensitive to this HP than standard approaches are to $\beta$. This means, showing performance of the algorithms with respect to different values of these hyperparameters for a wide enough range.

### 3. Difficult to read, the motivation is not immediately clear, many terms used vaguely without definition etc.
- please see clarity section below

### 4. The empirical gains are very limited
- from Figure 3, it looks like DEO is actually worse than Rainbow on a few hard-exploration tasks (e.g. MontezumaRevenge, Pitfall etc.) which is surprising and seems to contradict the paper's claims that DEO can better benefit from different types of exploration (which are likely to be most useful in hard-exploration environments). Can you include a more detailed discussion about this?
- it looks like DEO is unstable on PrivateEye. Do you understand why? This is somewhat worrisome and warrants more analysis and discussion.

### 5. The method is a simple extension of prior work (but this is a minor issue and not the main reason for my score)
- I suggest outlining more clearly whether there are any substantial technical differences between DEO and EO. it's also ok if there are no major modifications, but I think it's worth making this more clear.

# References
[1] Curiosity-driven Exploration by Self-supervised Prediction, Pathak et al. 2017

[2] Unifying Count-Based Exploration and Intrinsic Motivation, Bellemare et al. 2016

[3] Noisy Networks for Exploration, Fortunato et al. 2017

[4] Deep Exploration via Bootstrapped DQN, Osband et al. 2016

[5] Temporally Extended $\epsilon$-greedy exploration, Dabney et al. 2020


**Summary Of The Paper:**

This paper proposes an exploration approach based on learning multiple options, each corresponding to different intrinsic rewards. This is in contrast with the typical way of combining multiple intrinsic rewards which uses a weighted sum. The method is referred to as Deep Explore Options (DeepEO / DEO) and is an extension of Explore Options to the deep learning setting. The authors claim that DEO is more robust to ineffective or harmful intrinsic rewards, so it can be applied to a broader range of tasks without the need to tailor the set of intrinsic rewards for each task. The approach is evaluated on Atari and compared with a few baselines and ablations.

**Summary Of The Review:**

Overall, while the paper proposes a simple, intuitive approach and aims to solve an important problem, the experiments fall way short of supporting the claims of the paper. Many important baselines and ablations are missing. The paper is also quite difficult to read and the motivation isn't immediately clear. I think it also tries to make too many claims with some of them out-of-scope for the current work. I suggest focusing on a few core claims, highlighting the problem you are aiming to solve (i.e. clearly lay out the limitations of prior work and how precisely your approach addresses them), and making sure they are well-supported by the experiments.

---

> ### Author Response · Authors · 2022-11-18
> **Thank you for the in-depth review**
>
> We would like to thank the reviewer for the in-depth comment.
>
> We will take most of the feedback into account in a future version of the work. Below we address what can already be said
> - DeepEO is compared with Rainbow+RND+IO+Random in Appendix Figure 6, capturing both scalability and robustness. It is clear there that a weighted sum does not have these properties. In Campos et al. (2021), they cover transfer experiments essentially identical to a DeepEO setting, while noting that "Bagot et al. (2020) augments an agent with the ability to utilize another policy, which is learned in tandem based on an intrinsic reward function. This promising direction is complementary to our work, as it handles the case wherein there is no unsupervised pre-training phase." The Exploiting behavior can be seen in the performance of Pitfall and Freeway where the agent, when unconstrained to use the option use eps-greedy, performs better.
> - the comment on policy-based methods is very interesting as we have considered several times to use them instead of value-based methods, with the same intuition. The main issue was that the benchmark paper we use was the only extensive study and comparison of intrinsic motivation methods at the time, therefore we were constrained to value-based methods on Atari.
> - to clarify, there are indeed no major differences between EOs and DeepEOs beyond auxiliary task learning, the experience replay management and the adapted eps-greedy scheduling.
>
> The rest of the comments will be incorporated or clarified in a future version of the work. We deeply thank the reviewer for the detailed review.

---

> > ### Comment · Reviewer_u5M2 · 2022-11-24
> > **Post-Rebuttal Response**
> >
> > Thank you for responding to some of my concerns. Given that no major changes have been made to the paper, I will keep my score but I encourage the authors to take the feedback into account and submit the paper to a future venue.

---

### Official Review · Reviewer_xbC8 · 2022-10-26

**Confidence:** 4
**Correctness:** 3
**Technical Novelty And Significance:** 1
**Empirical Novelty And Significance:** 1
**Recommendation:** 3

**Clarity, Quality, Novelty And Reproducibility:**

**Clarity:**
- How DeepEO works is not clearly explained. Figure 1 seems to suggest that the exploit option and explore options can all be invoked by some central mechanism. However, how such a mechanism is learned is not explained. In actuality, it appears that calling the explore options might be additional actions given to the exploit agent, but this is not explained. It is also not clear how the exploit agent would learn to invoke the explore options based on its own reward. The bottom of page 4 seems to suggest that the explore options are randomly invoked, rather than having the exploit policy learn to invoke them. If this were the case, it would reduce the impact of the method considerably. The authors should clarify these points significantly.
- The intro incorrectly states that a weighted sum IM approach could not learn from several IM rewards at once, but there is no reason that it could not.
- Some acronyms are used without being introduced, such as WS and PER.
- Figure 2 b) bottom row and the associated explanation is very unclear. The message this was intended to communicate did not come across.
- The text states the minigrid environment is fully observed. In this case, what is the exploration challenge here?
- Section 4.1 tests an ablation of DeepEOs (Deep ExploreOptions) without options. What does it mean to remove options from this approach? How would that work? This is not explained and very unclear.
- The organization of the paper could be significantly improved. For example, Section 6 outlines various desiderata for an IM-Inc method. Why not put this section *before* the experiments, to motivate the experiments you chose to run? (For example, if you believe an IM method should be robust, this motivates running the robustness analysis that you conducted).
- Similar to the point above, Section 6 is quite long and appears to stray from the topic, becoming more of a second related work section by 6.2. This could be consolidated with the related work and condensed. The detailed explanation of Successor Features is not that relevant here. Instead, much more space could be devoted to explaining the method in a way that is reproducible.

**Quality:**
- The case for learning explore/exploit options is not presented in the paper in a compelling way. The paper repeatedly refers back to the weighed sum of intrinsic reward approach as being problematic because we must learn a coefficient that trades off between an explore/exploit objective. Learning such a coefficient hyperparameter is not an issue and can be easily tuned with a hyperparameter search. This fact leads the authors to awkwardly justify in several places where they also learned a weighting parameter (such as $\eta$ for RND) that this should not be considered the same as in the weighted sum approach in IM. The reason that this is awkward is because it's the wrong argument. It's not that learning a weighted sum is hard, it's that trying to simultaneously explore and exploit is the wrong objective. Exploration and exploitation are distinct phases of learning, and should not be mixed together throughout training and in the final inference policy. Being able to *learn* when to explore and when to exploit is a compelling idea, but it is not well communicated in this paper.
- Why only focus on curiosity as the IM options? The intro seems to suggest that intrinsic motivation is entirely about curiosity and exploration, but that is not true; other IM methods such as empowerment are potentially quite useful as well. Why was empowerment not considered as a way to train one of the IM options in this method? This would allow the agent to decide when to explore, gain control of the environment (minimize entropy), and exploit the task specific reward. This would have given the agent a greater range of intrinsically motivated behaviors to deploy to learn the task. In the current version, all three explore options appear similar, and it is not clear why three were needed.

**Originality:**
- As mentioned above, the difference between this work and Campos et al. 2021 is not clear. Please justify this better.
- This paper https://arxiv.org/abs/2107.07394 uses a multi-agent approach to decompose exploration and exploitation into different policies, and would be worth citing.

**Strength And Weaknesses:**

A strength of the paper is that the idea is compelling. Learning both exploit and explore options and being able to intelligently switch between them is a more principled approach to the exploration/exploitation problem than combining incentives to simultaneously explore and exploit into one reward.

There are three main weaknesses of the paper:
1) Novelty: the central idea of the paper appears to have been explored previously in Explore Options (Bagot et al. 2020) (albeit not with deep learning), and Campos et al. (2021). Particularly with Campos et al., it appears to also use exploratory options which are called by an exploiter agent, so the contribution of the current paper over Campos is not clear. This should be more thoroughly explained.

2) Effectiveness/Impact: The results on Atari reveal that DeepEO does not work effectively compared to the Rainbow or Rainbox+RND baselines. It appears that DeepEO provides a significant performance improvement above the baselines in only 2/11 environments, and these are both "easy" exploration tasks as described by the authors. These results are not compelling.

3) Clarity: The actual method used for DeepEO is unclear, as I will explain below.

**Summary Of The Paper:**

The paper extends the idea of ExploreOptions to the deep RL setting. Four different sub-policies are trained off-policy on the same data: an Exploit policy, and 3 Explore policies, which each use a different intrinsic motivation (IM) method to incentivize exploration (e.g. including RND). Results are presented in a gridworld and Atari.

**Summary Of The Review:**

Overall, while the idea behind the paper is compelling, it has been explored in prior work and was not presented in a compelling way here. The results indicate the proposed technique is less effective than well-known existing methods, limiting the potential impact of the paper.

If the authors can clarify some points I may be willing to increase my score. Can you please address:
- How this work is different from Campos et al. 2021 and what additional contributions does it make?
- Is this a hierarchical method in which a central controller learns which option to invoke, or does the exploit option learn to invoke the others (in which case Figure 2 should be revised), or are the options randomly invoked?
- Why focus on three similar exploration IM methods to create the options and not include empowerment?

---

> ### Author Response · Authors · 2022-11-18
> **Thank you for the in-depth review**
>
> We would like to thank the reviewer for the in-depth comment.
>
> Regarding comments comparing the work with Campos et al., it is mentioned in their paper: "Bagot et al. (2020) augments an agent with the ability to utilize another policy, which is learned in tandem based on an intrinsic reward function. This promising direction is complementary to our work, as it handles the case wherein there is no unsupervised pre-training phase." Essentially, DeepEOs provide a straightforward framework to train the options that Campos et al. use for transfer.
>
> - Regarding how the option is used: it follows from both the option framework in general, and the original Explore Options work in particular. The Exploiter can decide to call the option, letting the Explorer interact for c_switch steps. The discounted extrinsic rewards gathered along the way provide a reward to assign to the option. This is explained in the final paragraph of Section 3.2, just before diving into the adaptation of eps-greedy to the DeepEO.
> - the beta parameter finetuning required for a sum of rewards explodes with the amount of rewards, hence it is not scalable. In addition, if an IM motivates going to the "right" and another motivates going to the "left", one of the behaviors or both will be lost in a weighted sum (behavior assimilation problem).
>
> The rest of the comments will be incorporated or clarified in a future version of the work.
> We deeply thank the reviewer for the detailed review

---

> > ### Comment · Reviewer_xbC8 · 2022-11-18
> > **Thank you for the response.**
> >
> > Thank you for the clarifications. I hope the comments will help in revising the paper.

---

### Official Review · Reviewer_dS6L · 2022-10-27

**Confidence:** 2
**Correctness:** 3
**Technical Novelty And Significance:** 3
**Empirical Novelty And Significance:** 2
**Recommendation:** 3

**Clarity, Quality, Novelty And Reproducibility:**

As noted above, I consider lack of clarity to be the primary weakness of this work. Along with that lack of clarity follows reproducibility concerns; missing details would make replicating the experiments quite challenging. To the best of my knowledge, this extension of exploration option to a function approximation setting has not been published before, and the authors have provided what I consider an original analysis of methods for incorporating intrinsic rewards into an extrinsically motivated agent.

I wanted to ask—Bagot et al. (2020) use the term 'exploration options'; is there a particular reason why you are changing the name to Explore Options? I initially wondered whether or not you were referring to the same concept.

**Strength And Weaknesses:**

The exploration options framework is a very interesting setup, and this extension of the framework to a function approximation setting is an obvious next step for exploring the framework further. This is a fairly ambitious conference paper, combining both the experimental extension of exploration options to DeepEO with the more philosophical exploration of methods for incorporating intrinsic rewards into reinforcement learning agents.

I considered the most prevalent weakness of this paper to be the clarity of the document. I have a number of questions coming from undefined terms and missing details.
1. "the types of behaviors that we can extract with IM" (p. 1), "DeepEOs can extract the exploiting behavior" (p. 2) → I'm not sure what it means to extract a behaviour; can you clarify?
2. "However, by design they only make sense in a function approximation setting … " (p. 2) ← I don't understand this last part of the paragraph at all.
2. "e.g. as a default behavior when no rewards are known." (p. 3) If the Exploiter's learning and decisions are based on extrinsic reward, how do you envision the Exploiter learning this kind of default behaviour? What would make it better, from the Exploiter's point of view, than choosing random actions, if "no rewards are known"? Or is this meant for a transfer learning setting?
3. "The only practical issue with such approaches that learn different agents is if the scarce relative interacting of all agents begets the necessity of Offline RL (Fujimoto et al., 2019; Levine et al., 2020))." (p. 3)← I don't understand why "scarce relative interacting of all agents" would beget "the necessity of Offline RL." I'm not sure what "relative interacting" is, for one thing.
4. "This type of architecture is quite common … but very rarely so when dealing with IM, mainly because the WS is hard to scale." (p. 3) ← When you say "this type of architecture," are you referring to architectures where the parameters of the model are shared across auxiliary tasks? This statement would be more helpful with more explanation of why scaling the WS causes particular problems in this setting beyond the usual parameter sensitivity issues associated with WS.
5. I think the j in the loss combination function (p. 3) is supposed to be a super- or subscript—or else I'm not sure why you're multiplying by j, which is how I would read that. Also, I couldn't find definitions for your $\mathcal L$s prior to this point. Why doesn't this equation match line 7 of Algorithm 1?
6. I don't understand what you mean by "environment-aware" (p. 3)
7. Can you provide more intuition for why the individual loss functions don't have to be balanced in the combined loss function?
8. "a unique buffer" (p. 4) ← What do you mean by a unique buffer? What is it unique in comparison to?
9. "Instead, we note that ϵ-greedy means allocating a fraction ϵ of the agent’s time to exploring. In order to preserve this idea, we enforce our Explorer agents to occupy ϵ/2 of the agent’s time, while our Exploiter otherwise follows an ϵ/2-greedy exploration strategy." (p. 4) ← Can you provide more detail about how you achieved this scheduling? This detail could go in an appendix if you're worried about space requirements.
10. What is a "coverage map"? (p. 4)
11. "Except for Solaris, the Exploiter always improves on DeepEO where option usage is forced." (p. 6) ← I might need your help to understand the figures this way. From my reading, there is so much variance in the performance of both Exploiter and DeepEO that it doesn't seem necessarily true that the Exploiter improves on DeepEO in any environment other than Pitfall and Freeway. What am I missing in my reading of the figures?
12. "we always improve on the lower bound." (p. 6) ← I don't understand what this means.

And then I also have a list of parts of the paper that can be made sense of having read the whole document, but based on whether they fall in the paper, they need to be better explained. This list is not exhaustive.
- "relying on a single signal to help exploitation." (p. 1)← I found this confusing; does "single signal" refer to r_t? This is particularly difficult to follow being so early in the paper where you haven't introduced any alternative.
- Please be explicit about whether "left" and "right" actions turn the agent or move the agent to a new cell, or something else?
- Please explain the extrinsic reward function for your four-room environment
- "We also use it to train … an untouched DQN agent" (p. 5) ← My best guess about what "it" is in this sentence is the first-cell-visit intrinsic reward, so became quite lost about the difference between the DQN+IM agent and the "untouched DQN agent."

One part of the paper that I would consider an error is the treatment of the Noisy TV problem in Section 3.2. As it is described, it seems like the intention is for the third intrinsic reward function to simulate the Noisy TV problem. However, instead of setting up a reward such that one part of the environment will consistently present as a distractor generating high intrinsic reward (which is true of the "noisy TV"), this reward appears to be uniformly variable across each environment, which does not seem to appropriately simulate the proble.

I enjoyed Section 6.2; it wasn't completely clear, but I found it a helpful overview of existing IM-Inc approaches. I also appreciated the use of the Taïga et al. (2019) benchmark to contextualize DeepEO with other IM methods.

A few slight wording issues that can be read as inaccurate:
- "The agent is therefore deeply tied to the reward signal, and tends to fail when said signal is sparse or noisy." (p. 1) ← I suggest rephrasing; while this type of failure is indeed typical of present RL agents, this phrasing makes it sound like the problem is inherent to the RL framework.
- "IM biologically refers to the natural tendency of organisms to explore."  (p. 1) ← This characterism isn't necessarily accurate. One of the classical definitions of intrinsic motivation is 'engagement in an activity for its own sake,' (Deci, 1975, p. 23) which doesn't perfectly align with the way the word 'explore' is typically used, especially in the context of reinforcement learning.
- "i.e. additional actions," (p. 1) ← It is a bit misleading to simply call options additional actions, as they have a very different structure from standard actions (I understand that they are chosen by the Exploiter just like low-level actions) so I'd recommend rephrasing.
- "to explore for a fixed amount of time." (p. 1) ← You may want to avoid making it sound like the 'fixed time' part is inherent to your definition of exploration options; while the prototype implementation of exploration options provided by Bagot et al. (2020, p. 4) does indeed only use them to explore for a fixed amount of time, they imply that the theoretical architecture could support more general termination conditions.
- "Motivating exploration is by far the most common goal then, and is divided into…" (p. 7) ← While I agree that this is the division that Aubret et al. (2019) make, the phrasing makes it sound like it is the only possible division, when there are other works that have used other distinctions to taxonomize intrinsic rewards (e.g. Oudeyer & Kaplan, 2009)

Typos and grammatical suggestions (no intended influence on score)
- "to the a weighted" (p. 1)→ "to a weighted"
- The replay buffer in Figure 1 could probably use some more ellipses, ideally, like "s, a, r^e, r^{i1}, r^{i2}, … , s' "
- "the Explore Option" (p. 2) → might make more sense as plural; aren't there multiple Explore Options?
- "the IM rewards themselves guiding" (p. 3) → "the IM rewards guiding"
- "3 majors improvements" (p. 4) → "3 major improvements"
- "less-visited states, that" (p. 4) → "less-visited states that" or "less-visited states, which"
- "explicited" (p. 4) → explicit is an adjective, rather than a verb; maybe you want "shown" or something similar
- "we evaluate our Exploiter" (p. 5) → "we also evaluate our Exploiter" (to help make it clear that you are also evaluating DeepEO while retaining the options—in fact, adding an explicit sentence noting that you are including DeepEO in the evaluations wouldn't go amiss!)
- "Guillaume Lample and Devendra Singh Chaplot. Playing fps games …" (p. 11) → "FPS games"
- "and lead to" (p. 14) → "and led to"


Deci, E.L. (1975). Conceptualizations of Intrinsic Motivation. In: Intrinsic Motivation. Springer, Boston, MA. doi:10.1007/978-1-4613-4446-9_2

**Summary Of The Paper:**

This paper offers a straightforward extension of the exploration options framework (Bagot et al., 2020)—which was initially introduced as a prototype for tabular environments—to a deep-learning, function approximation setting. The method is empirically evaluated in the Atari Learning Environment, following a standardized benchmark setup. The paper further considers a broader question, "How should we incorporate intrinsic reward signals into learning algorithms?" The authors, referring to methods that aim to answer this question as IM Incorporation methods, distill a set of normative properties that they recommend for future methods of this type.

**Summary Of The Review:**

I am recommending the rejection of this paper because it is insufficiently clear and poorly organized. Important details are often left implicit, so I often felt like I had to figure out the details via guesswork. I only found one concerning error, but because I had such difficulty understanding the work, I am not confident in being able to assess the work completely. I believe the research the authors set out to present uses fairly appropriate choices of experiments and sets out interesting recommendations for future work but the presentation itself isn't strong enough for publication at this venue.

---

> ### Author Response · Authors · 2022-11-18
> **Thank you for the in-depth review**
>
> We would like to thank the reviewer for the in-depth comment.
>
> We will be addressing below some of the main points and questions risen (within space constraints):
> 1. "`extracting` behaviors" - IM rewards could be simply said to "generate" behaviors. However, in the context of the paper, we compare DeepEOs with a weighted sum. A WS combines the reward signals into a single policy that balances exploration and exploitation, but effectively loses its ability to do either independently. In comparison, DeepEOs combine rewards while still being able to "extract" (use) the purely exploring/exploiting behaviors from the combined policy.
> 2 and 3. the Exploiter can observe that the option leads to a higher average return than any other action in most of the state-action space. Through function approximation, it can therefore affect a higher value to the option in unexplored parts of the state-action space, where no rewards are known, and use the option as a "default" there. The resulting behavior is similar to having the option optimistically initialized, though this is achieved through learning.
> 4. Sorry for the lack of clarity. When the agent learns offline, it leads to a distributional shift where the training data is vastly different from testing data. This can also appear if the agent is not allowed to interact much with the environment, and most of its training data does not come from its policy (as would happen with a lot of Explorers in DeepEOs). Offline/Batch RL aims to solve this distributional shift.
> 6. the losses refer to any RL Agent loss. The IM-Gen modules are independent from the main architecture so their loss is irrelevant to the agents.
> 9. a straightforward approach to DeepEOs would be to use separate buffers for each agent. This would mitigate distributional shift but completely drop the benefits of off-policy learning, mainly preventing the Exploiter from observing Explorers
>
> The rest of the comments will be incorporated or clarified in a future version of the work.
>
> Regarding the NoisyTV problem, our objective here was rather to point out that it is possible that a reward function becomes harmful to the agent for a given task and environment (in the sense that optimizing it does not lead to optimal policies for our task). The random reward does not try to emulate the NoisyTV problem specifically, but any disruptive reward.
>
> The change from Exploration to Explore in the method name was discussed directly with the author following their own shift in taxonomy, but not changed in the original work.
>
> We deeply thank the reviewer for the detailed comments on clarity and wording, and we will take those into account in future versions of the work.

---

### Decision · Program_Chairs · 2023-01-20

**Decision:**

Reject

**Justification For Why Not Higher Score:**

The paper is unanimously rejected by the reviewers with very similar concerns. It is hard to think about overriding this recommendation.

**Justification For Why Not Lower Score:**

N/A.

**Metareview: Summary, Strengths And Weaknesses:**

This paper, among other things, proposed the idea of scaling up the options-based exploration to the deep RL setting. The previous paper introduced such ideas only in the tabular case. The method is evaluated in Atari 2600 games.

Overall, the reviewers did agree this was a compelling idea. Different reviewers raised different concerns, such as the paper being a simple extension of previous work. While it is debatable this is grounds for rejection, there is one clear reason this paper is not ready: all reviewers unanimously agreed that the document has several issues regarding its clarity, with undefined terms, missing details, poorly presented motivation, and so on. Given that and the concern raised by the reviewers that multiple claims are not well-supported by the experiments I’m recommending the rejection of this paper. I suggest the authors to go over the reviewers comments when preparing a new version of the manuscript to address some of their concerns.; this will greatly improve the paper.


**Summary Of Ac-Reviewer Meeting:**

Not applicable